# Basis Sharing: Cross-Layer Parameter Sharing for Large Language Model Compression

**Jingcun Wang**
Technical University of Darmstadt
`jingcun.wang@tu-darmstadt.de`

**Yu-Guang Chen**
National Central University
`andyygchen@ee.ncu.edu.tw`

**Ing-Chao Lin**
National Cheng Kung University
`iclin@csie.ncku.edu.tw`

**Bing Li**
University of Siegen
`Bing.Li@uni-siegen.de`

**Grace Li Zhang**
Technical University of Darmstadt
`grace.zhang@tu-darmstadt.de`

## Abstract

Large Language Models (LLMs) have achieved remarkable breakthroughs. However, the huge number of parameters in LLMs require significant amount of memory storage in inference, which prevents their practical deployment in many applications. To reduce memory storage of LLMs, singular value decomposition (SVD) provides a promising solution to approximate weight matrices for compressing LLMs. In this paper, we take a step further to explore parameter sharing across different layers with SVD to achieve more effective compression for LLMs. Specifically, weight matrices in different layers are decomposed and represented as a linear combination of a set of shared basis vectors and unique coefficients. The types of weight matrices and the layer selection for basis sharing are examined when compressing LLMs to maintain the performance. Comprehensive experiments demonstrate that Basis Sharing outperforms state-of-the-art SVD-based compression approaches and parameter sharing techniques, especially under large compression ratios.

## 1 Introduction

Large Language Models (LLMs) have revolutionized natural language processing by enabling machines to understand human language more accurately. Although these models have remarkable capabilities, they are computation- and memory-intensive, making their deployment on resource-constrained devices challenging. To address this challenge, model compression has become a widely adopted technique to reduce model size and complexity.

Common compression techniques, such as model distillation (Gu et al., 2024; Magister et al., 2023; Jiang et al., 2023b; Huang et al., 2022; Qiu et al., 2024), pruning (Frantar & Alistarh, 2023; 2022; Ma et al., 2023; Sun et al., 2024; Jiang et al., 2024; Petri et al., 2023), and quantization (Lin et al., 2024; Zhao et al., 2024; Ashkboos et al., 2024; Xiao et al., 2023; Sun et al., 2023), early-exit (Chen et al., 2024; Wang et al., 2024a), etc. have been extensively studied. While such techniques are effective in many scenarios, these methods often require hardware modification and expensive retraining. Compression techniques based on low-rank approximation with, e.g., Singular Value Decomposition (SVD) (Yuan et al., 2023; Hsu et al., 2022; Wang et al., 2024b), provide a promising alternative since they are not restricted by such constraints. In SVD-based weight compression, a weight matrix in a layer is processed individually by decomposing it into three matrices. By removing small singular values in the decomposed diagonal matrix, the original weight matrix can be approximated with fewer number of weight values.

Despite the benefits of SVD-based weight compression, the potential of grouping layers for weight approximation and compression has not been explored thoroughly. Since weight matrices in different layers of an LLM might share similarity, parameter sharing across layers can be exploited to further compress weight matrices for LLMs. In sharing parameters across layers, Hay & Wolf (2024) trained a small language model by restricting weight matrices in some layers to be the same. On the one hand, this brute-force method leads to significant performance degradation since weight matrices in different layers should vary to maintain their functionalities. On the other hand, it is impractical to train LLMs from scratch due to limited training data or high training costs.

Contrary to previous work, in this paper, we use pretrained LLMs to enable weight matrices across layers to share a common set of basis vectors but still retain their different functionalities with unique coefficients. Our method, called Basis Sharing, can compress LLMs effectively. In summary, our contributions are as follows:

1. We propose to represent weight matrices across different layers in a pretrained LLM with a linear combination of a set of shared basis vectors and coefficients unique to specific layers. This basis sharing can effectively reduce the number of parameters in LLMs while only affecting the performance of LLMs slightly.

2. We examine cross-layer basis sharing for different types of weight matrices in LLMs according to the incurred compression errors. The types of weight matrices whose sharing across layers does not incur significant compression error are selected for compressing LLMs.

3. For the selected types of weight matrices, we also develop a criterion to group layers to share a set of basis vectors but have individual coefficients to preserve the performance of LLMs.

4. We conduct extensive experiments on a variety of LLMs, including the LLaMA family (Touvron et al., 2023a;b), OPT-6.7B (Zhang et al., 2022), Mistral-7B (Jiang et al., 2023a), and GPT-2 (Radford et al., 2019). Our Basis Sharing can surpasses the state-of-the-art SVD-based methods in both generation tasks and downstream reasoning tasks without any fine-tuning under compression ratios from 20% to 50%. Specifically, compared with state-of-the-art SVD-based compression approaches, Basis Sharing can further reduce the perplexity by up to 25% on generation tasks and improve accuracy by up to 4% on downstream reasoning tasks under the same compression ratio.

## 2 RELATED WORK

**Large Language Model Compression**   LLM compression techniques include model distillation, pruning and quantization, etc. Gu et al. (2024); Huang et al. (2022); Magister et al. (2023); Jiang et al. (2023b) successfully applied model distillation to LLM by retraining, which incurs high computational cost. Frantar & Alistarh (2023; 2022); Sun et al. (2024); Ma et al. (2023) pruned weights that are less sensitive to outliers. However, the resulting unstructured weight matrices do not provide meaningful compression benefits on real hardware. Structured pruning techniques, such as 2:4 or 4:8 pruning, can achieve effective compression but restrict a fixed 50% pruning ratio, which limits flexibility in balancing performance and compression ratio. Zhao et al. (2024); Ashkboos et al. (2024); Lin et al. (2024); Xiao et al. (2023) allocated higher quantization bits to weights with larger influence on outliers, but it does not reduce the number of parameters, limiting its impact on overall compression.

**SVD-based Weight Compression**   SVD-based weight compression has a flexible compression ratio to maintain performance without retraining. Golub et al. (1987) were the first to apply SVD for neural network compression, and Lv et al. (2023); Wu et al. (2023) extended this approach to shallow transformer models (Vaswani, 2017). However, in LLM compression, these methods incur significant errors since they do not consider outliers in activations. FWSVD (Hsu et al., 2022) addresses this issue by incorporating the impact of outliers through the Fisher information analysis of weight matrices. However, this method requires gradient information during training process, which is computationally prohibitive for LLMs. ASVD (Yuan et al., 2023) alleviates this problem by selecting key channels in the weight matrix based on their sensitivity to outliers and minimizing compression error in these channels. While it avoids the need for gradients, ASVD still lacks a direct connection

between SVD truncation error and the overall model compression error. SVD-LLM (Wang et al., 2024b) improves this by introducing a whitening matrix that captures outlier information, effectively reducing compression error. *However, all of these methods focus only on compressing individual weight matrices within a single layer, missing the opportunity to exploit weight compression across multiple layers.*

**Parameter Sharing**  Parameter sharing reduces model size by reusing weight matrices across different layers. Inspired by recurrent neural networks, Dehghani et al. (2019) explored this concept within transformers by restricting all layers in the encoder and decoder to share the same weights. Similarly, Reid et al. (2021) divided transformer parameters into two groups (attention-related and feedforward-related) and compressed the model by sharing weights within each group. Takase & Kiyono (2021) applied selective weight sharing, where specific layers shared the same weights rather than all layers. Beyond direct weight sharing, Xiao et al. (2019); Bhojanapalli et al. (2021) introduced the idea of sharing attention scores between layers. By reusing attention scores, some weight matrices for attention computation could be discarded. Dynamic Tying (Hay & Wolf, 2024) determines layer-wise weight sharing during training using reinforcement learning, which is still time-consuming for large LLMs. *All of these approaches have been tested only on smaller transformer models and typically require training from scratch or full parameter fine-tuning, which makes them impractical for LLMs.*

## 3  METHODOLOGY

Contrary to the previous techniques that require training from scratch and weights in some layers are restricted to be the same during training, we adopt a pretrained LLM to explore representing weights across different layers with combinations of a set of shared basis vectors and individual coefficients. Since the set of basis vectors can be shared across several layers, the number of parameters in the LLM can thus be reduced effectively. The difference between the previous weight sharing method and our Basis Sharing is illustrated in Figure 1.

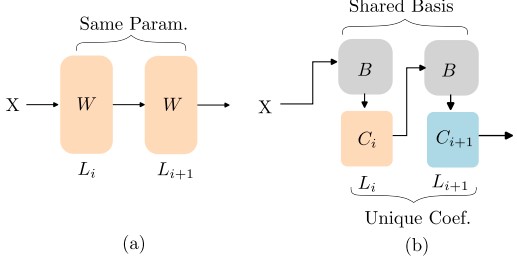

Figure 1: (a) Two layers share the same weight matrix in previous work. (b) Two layers share the same basis matrix but have their individual coefficients in our work.

To exploit the cross-layer parameter sharing to compress LLMs, the subsequent subsections address the following challenges: 1) What methodologies can be used to process the weight matrices across layers in an LLM to determine a set of shared basis vectors and individual coefficients? 2) Which types of weight matrices across layers in an LLM can take advantage of parameter sharing without affecting its performance significantly? 3) Which layers can share a set of basis vectors in an LLM without affecting its performance significantly?

### 3.1  REPRESENTING WEIGHT MATRICES ACROSS LAYERS WITH COMBINATIONS OF BASIS VECTORS AND COEFFICIENTS

Suppose that we have weight matrices across $n$ layers, denoted as $\boldsymbol{W}^{(1)} \ldots \boldsymbol{W}^{(n)}, \boldsymbol{W}^{(i)} \in \mathbb{R}^{d_1 \times d_2}$. To derive a set of shared basis vectors and coefficients to represent such weight matrices, intuitively, such matrices can be horizontally concatenated into one matrix, denoted as $\boldsymbol{W} \in \mathbb{R}^{d_1 \times nd_2}$, and singular value decomposition (SVD) can be applied to decompose this matrix into three matrices: $\boldsymbol{U}, \boldsymbol{\Sigma}, \boldsymbol{V}^T$. $\boldsymbol{\Sigma}$ is a $d_1 \times nd_2$ diagonal matrix consisting of singular values of $\boldsymbol{W}$.

By selecting the top $k$ singular values in $\boldsymbol{\Sigma}$, $\boldsymbol{W}$ can be approximated as $\boldsymbol{W} \approx \boldsymbol{W}_k = \boldsymbol{U}_k \boldsymbol{\Sigma}_k \boldsymbol{V}_k^T$, where the dimensions of $\boldsymbol{U}_k, \boldsymbol{\Sigma}_k$ and $\boldsymbol{V}_k^T$ are $d_1 \times k$, $k \times k$, and $k \times nd_2$, respectively. The value of $k$ should be determined to balance the compression ratio and the performance of the compressed LLM (Appendix A.2 shows the evaluation of $k$ under a given compression ratio). $\boldsymbol{W}_k$ can be rewritten as $\boldsymbol{W}_k = \boldsymbol{B} \boldsymbol{V}_k^T$, where $\boldsymbol{B}$ is the multiplication result of $\boldsymbol{U}_k$ and $\boldsymbol{\Sigma}_k$. We call $\boldsymbol{B}$ a basis matrix and a column of $\boldsymbol{B}$ is a basis vector, denoted as $\boldsymbol{B}_{:,i}$. $\boldsymbol{V}_k^T$ can be considered as a coefficient matrix, i.e., $\boldsymbol{V}_k^T = \boldsymbol{C}$. Accordingly, the $j^{th}$ column of the original weight matrix $\boldsymbol{W}^{(i)}$ in the $i^{th}$ layer can be

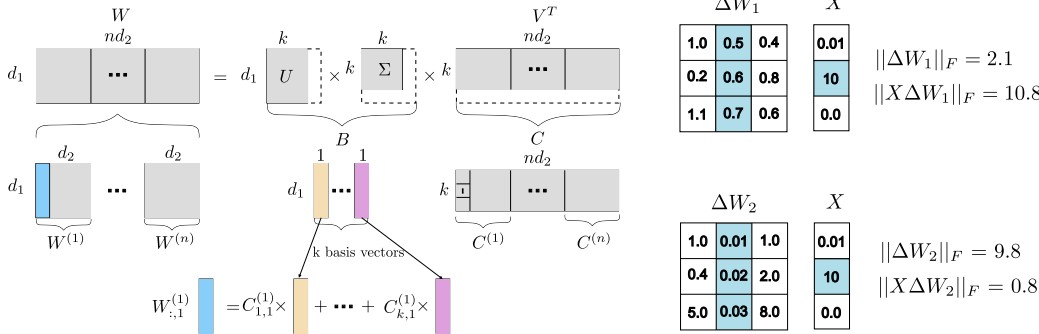

Figure 2: Weight matrices across $n$ layers are concatenated horizontally into a weight matrix, which is processed by SVD. The $j^{th}$ column of the original weight matrix in a layer can be represented as a linear combination of $k$ shared basis vectors and coefficients.

Figure 3: $\Delta W_1$ and $\Delta W_2$ are differences with respect to the original weight matrix after compression. $||\Delta W_1||_F$ is smaller than $||\Delta W_2||_F$, but $||X\Delta W_1||_F$ is larger than $||X\Delta W_2||_F$.

approximated as a liner combination of $k$ basis vectors and individual coefficients as follows.

$$W_{:,j}^{(i)} \approx \sum_{m=1}^{k} B_{:,m} C_{m,j}^{(i)}. \tag{1}$$

where $C^{(i)}$ is the coefficient matrix in $i^{th}$ layer. The process of weight matrix approximation and representation is illustrated in Figure 2.

In the weight matrix approximation with SVD above, input data, denoted as $X$, are not considered. In fact, the result of $XW$ instead of $W$ is used in inference. Accordingly, applying SVD directly onto weight matrices without incorporating input data might lead to significant computation loss and potentially affect the performance of the LLM. Figure 3 illustrates an example, where a weight matrix approximated with SVD leads to a large compression loss in the form of Frobenius loss, denoted as $||X\Delta W||_F$. Since the second element in the input data affects the computation accuracy significantly, the second column of the weight matrix should be approximated more accurately compared with other columns to reduce the overall computation loss. Yuan et al. (2023); Wang et al. (2024b) also pointed out similar results.

To incorporate the effect of input data into the weight approximation with SVD to maintain the performance of the LLM, we will scale the concatenated weight matrix $W$ with a matrix $S \in \mathbb{R}^{d_1 \times d_1}$ as follows

$$W = S^{-1} SW = S^{-1}(SW). \tag{2}$$

The matrix $S$ should be evaluated to represent the impact of input data on the weights, so that it can adjust $W$ accordingly to reflect the significance of different input data. To obtain appropriate $S$, we will adapt the techniques developed in Wang et al. (2024b), where $S$ can be evaluated with $S(S)^T = cholesky((X)^T X)$. However, $X$ in their technique refers to input data of a layer instead of several layers in our method. To evaluate $S$ considering several layers, we will vertically concatenate the input matrices in such layers, denoted as, $X^{(1)}, \ldots, X^{(n)}$, and compute the $S$ with the concatenated $X$. In our experiments, we use 256 samples from WikiText-2 (Merity et al., 2016) with each 2048 tokens to evaluate $X$, similar to that in Wang et al. (2024b).

Instead of applying SVD directly on the concatenated weight matrix $W$, we will decompose $SW$ with SVD and approximate this scaled weight matrix $SW \approx U'_k \Sigma'_k V'_k = B'C'$, where $B'$ and $C'$ are the revised basis matrix and coefficient matrix, respectively. To recover the approximated weight matrix for computation in inference, $S^{-1}$ will be multiplied with $B'$, the result of which will be the final adjusted basis matrix, i.e.,

$$W \approx S^{-1} U'_k \Sigma'_k V'_k = S^{-1} B'C' = B''C', \tag{3}$$

where $B''$ is the final adjusted basis matrix in our paper.

## 3.2 SELECTION OF WEIGHT MATRICES IN LLMs FOR CROSS-LAYER PARAMETER SHARING

Modern LLMs are constructed based on the decoder-only transformer architecture. A layer in such an architecture includes several types of weight matrices, which have different functions. $\boldsymbol{W}_K$, $\boldsymbol{W}_Q$ and $\boldsymbol{W}_V$ are three types of projection matrices, which are used to generate the key, the query and the value matrices. $\boldsymbol{W}_O$, another type of weight matrices, further transforms the attention result to build a new representation for an input embedding. $\boldsymbol{W}_{Up}$ and $\boldsymbol{W}_{Gate}$(used in LLaMA and LLaMA2), further types of weight matrices, represent this transformation result into a high-dimension embedding. Afterwards, $\boldsymbol{W}_{Down}$, the last type of weight matrices, projects the high dimension embedding back to the low dimension embedding. The types of weight matrices above have different functions, so that we need to determine which type of weight matrices can take advantage of cross-layer basis sharing with SVD described in Section 3.1 without affecting the performance of the LLM significantly.

First of all, the type of matrices whose function are to project a high-dimension embedding into a low-dimension embedding such as $\boldsymbol{W}_{Down}$ cannot take advantage of the cross-layer parameter sharing. The reason is that after the horizontal concatenation of such matrices, the rank of the concatenated matrix will be larger than that of an individual matrix. Under the same compression ratio, compressing the concatenated matrix with SVD incurs a larger Frobenius loss than the original weight matrix.

For the remaining types of weight matrices including $\boldsymbol{W}_K$, $\boldsymbol{W}_Q$, $\boldsymbol{W}_V$, $\boldsymbol{W}_O$, $\boldsymbol{W}_{Up}$ and $\boldsymbol{W}_{Gate}$, we will determine whether each of them can use cross-layer basis sharing by examining the Frobenius loss resulted from this sharing. To explain this concept, we use basis sharing across two layers for $\boldsymbol{W}_K$ in LLaMA2-7B as an example. Assume that we remove small singular values by applying SVD on $\boldsymbol{S}_K^{(i)}\boldsymbol{W}_K^{(i)}$ to achieve a compression ratio of 20%, where $\boldsymbol{W}_K^{(i)}$ is $\boldsymbol{W}_K$ matrix in the $i^{th}$ layer ($i \in [1, 32]$) and $\boldsymbol{S}_K^{(i)}$ is the corresponding $\boldsymbol{S}$ matrix for $\boldsymbol{W}_K^{(i)}$. The resulting Frobenius loss of each layer under this compression ratio will be evaluated. To evaluate the Frobenius loss incurred by basis sharing, we horizontally concatenate $\boldsymbol{W}_K^{(i)}$ of the $i^{th}$ layer and $\boldsymbol{W}_K^{(j)}$ of the $j^{th}$ layer as $\boldsymbol{W}_K^{(i,j)}$ where $j \neq i$, $i, j \in [1, 32]$. SVD is applied on $\boldsymbol{S}_K^{(i,j)}\boldsymbol{W}_K^{(i,j)}$ to remove small singular values to achieve the same compression ratio, where $\boldsymbol{S}_K^{(i,j)}$ is the corresponding S matrix for $\boldsymbol{W}_K^{(i,j)}$. Afterwards, we evaluate the incurred Frobenius loss of basis sharing across two layers. Similarly, we repeat the process above for $\boldsymbol{W}_O$. The results are illustrated in Figure 4, where the number/color in a block represents the resulting Frobenius loss if a basis matrix is shared between two layers and the numbers in the diagonal direction are obtained by applying SVD to the scaled weight matrix of a layer directly.

Figure 4 compares the results of basis sharing for $\boldsymbol{W}_K$ and $\boldsymbol{W}_O$. Basis sharing across two layers for $\boldsymbol{W}_K$ can reduce the Frobenius loss. For example, when SVD is applied on $\boldsymbol{S}_K\boldsymbol{W}_K$ for the $9^{th}$ and $10^{th}$ layers separately, the resulting Frobenius loss is evaluated as $33508.2 + 33174.7 = 66682.9$. When the $9^{th}$ and $10^{th}$ layers share a common basis matrix, the Frobenius loss resulting from compression becomes smaller, i.e, $61817.3 < 66682.9$. This indicates that allowing parameter sharing across two layers for $\boldsymbol{W}_K$ can enhance computation accuracy. This trend can be seen in $\boldsymbol{W}_K$, $\boldsymbol{W}_Q$, $\boldsymbol{W}_V$, $\boldsymbol{W}_{Up}$ and $\boldsymbol{W}_{Gate}$ (Appendix A.8 show the results). Accordingly, basis sharing across layers can be applied on such matrices.

On the contrary, basis sharing for $\boldsymbol{W}_O$ in $9^{th}$ and $10^{th}$ layers incurs the increase of the Frobenius loss, i.e., $10618.3 > 4355.1 + 4895.7$. Accordingly, this parameter sharing should not be applied on $\boldsymbol{W}_O$ to avoid significant computation loss. For such matrices, we will apply SVD to process the individual matrix in each layer separately.

## 3.3 SELECTION OF LAYERS FOR BASIS SHARING

Section 3.1 determines which types of weight matrices can be shared across layers. This subsection then determines which layers can share basis vectors to represent such types of weight matrices. To select layers for basis sharing, the basis sharing of such layers should not incur Frobenius loss larger than without sharing. According to Figure 4, the group of two adjacent layers leads to smaller Frobenius loss than the sum of the Frobenius loss of two separate layers. Based on this analysis, we will group adjacent layers with the order from the first layer to the last layer. Take a group of two layers as an example. The first layer and the second layer are grouped for basis sharing, followed by the group of the third layer and the fourth layer, etc.

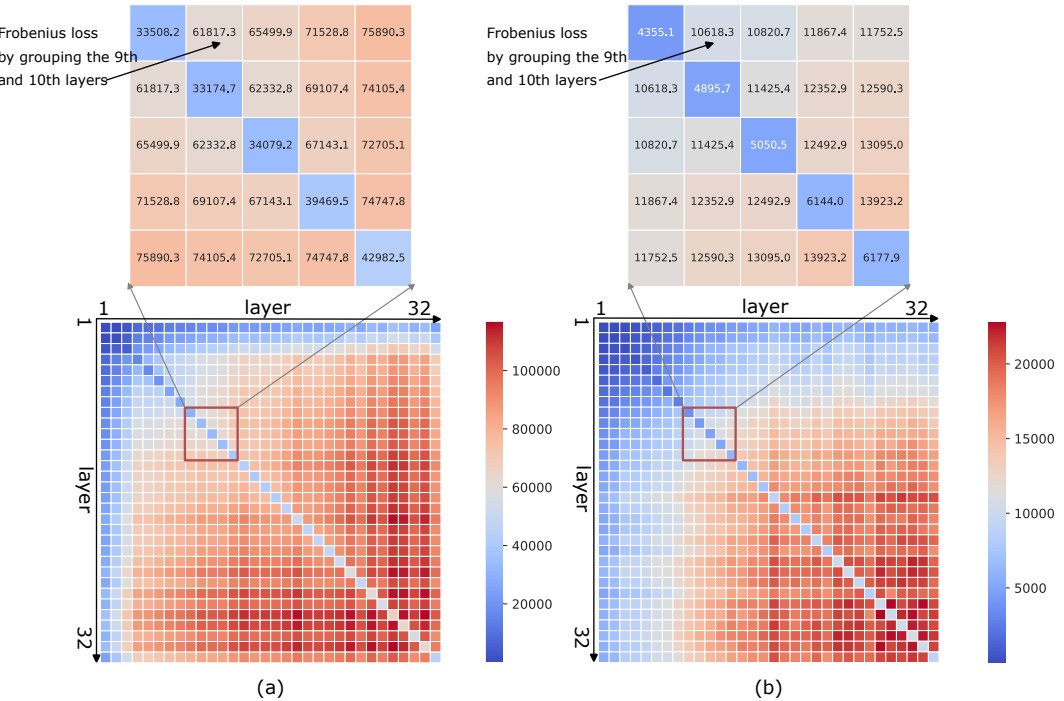

Figure 4: Frobenius loss incurred by basis sharing across any two layers. The number/color in a block represents the resulting Frobenius loss if a basis matrix is shared by two layers and the numbers in the diagonal direction are obtained by applying SVD to the scaled weight matrix of a layer directly. (a) Frobenius loss incurred by basis sharing across two layers for $W_K$ in LLaMA2-7B. (b) Frobenius loss incurred by basis sharing across two layers for $W_O$ in LLaMA2-7B.

## 4 EXPERIMENTS

### 4.1 SETTINGS

**Baseline** We compare with the work where SVD-based weight approximation in each individual layer is applied without cross-layer parameter sharing. Such work includes ASVD (Yuan et al., 2023), FWSVD (Hsu et al., 2022) and SVD-LLM (Wang et al., 2024b). We also compared our method with Dynamic Tying (Hay & Wolf, 2024), where weights in some layers are restricted to be the same by training from scratch. Since this method can only be applied on small language models, only GPT2 (Radford et al., 2019) was used to compared our method and Dynamic Tying.

**Models and Datasets.** We evaluate our method using several models. For LLMs, many models are evaluated, namely LLaMA family (LLaMA-7B, LLaMA-13B, LLaMA-30B, LLaMA2-7B) (Touvron et al., 2023a;b), OPT-6.7B (Zhang et al., 2022), Mistral-7B (Jiang et al., 2023a), GPT2. Three language modeling datasets used in our experiment include WikiText-2 (Merity et al., 2016), PTB (Marcus et al., 1993) and C4 (Raffel et al., 2019). Seven reasoning datasets used in the experiments include OpenbookQA (Banerjee et al., 2020), WinoGrande (Sakaguchi et al., 2021) HellaSwag (Zellers et al., 2019), PIQA (Bisk et al., 2020), MathQA (Amini et al., 2019), ARC-e, ARC-c (Clark et al., 2018). All the reasoning tasks are tested in zero-shot setting with the implementation of LM-Evaluation-Harness framework (Gao et al., 2024).

**Implementation details** All of our models are based on the model implemented by the Hugging Face. LLaMA-30B are implemented with FP16, the rest models are implemented with FP32. To evaluate $S$, FP64 is used to maintain the computation precision. All experiments are tested on two NVIDIA A100 80GB GPUs. $S$ is derived through 256 samples from WikiText-2 with 2048 sequence length. When the compression ratio is 40% or larger than 40% , the incurred compression errors increase, so that the output of a layer as the input of the next layer deviates significantly from its

Table 1: PPL(↓) and Zero-shot(↑) performance of LLaMA-7B with Basis Sharing and baselines under 20% to 50% compression ratio on three language modeling datasets and seven common sense reasoning datasets. The $S$ of all tasks is obtained with the dataset WikiText-2.

| RATIO | METHOD | WikiText-2↓ | PTB↓ | C4↓ | Openb. | ARC_e | WinoG. | HellaS. | ARC_c | PIQA | MathQA | Average↑ |
|---|---|---|---|---|---|---|---|---|---|---|---|---|
| 0% | Original | 5.68 | 8.35 | 7.34 | 0.28 | 0.67 | 0.67 | 0.56 | 0.38 | 0.78 | 0.27 | 0.52 |
| 20% | SVD | 20061 | 20306 | 18800 | 0.14 | 0.27 | 0.51 | 0.26 | 0.21 | 0.53 | 0.21 | 0.31 |
| | FWSVD | 1727 | 2152 | 1511 | 0.15 | 0.31 | 0.50 | 0.26 | 0.23 | 0.56 | 0.21 | 0.32 |
| | ASVD | 11.14 | 16.55 | 15.93 | 0.25 | 0.53 | 0.64 | 0.41 | 0.27 | 0.68 | 0.24 | 0.43 |
| | SVD-LLM | 7.94 | 18.05 | 15.93 | 0.22 | 0.58 | 0.63 | 0.43 | 0.29 | 0.69 | 0.24 | 0.44 |
| | Basis Sharing | 7.74 | 17.35 | 15.03 | 0.28 | 0.66 | 0.66 | 0.46 | 0.36 | 0.71 | 0.25 | 0.48 |
| 30% | SVD | 13103 | 17210 | 20871 | 0.13 | 0.26 | 0.51 | 0.26 | 0.21 | 0.54 | 0.22 | 0.30 |
| | FWSVD | 20127 | 11058 | 7240 | 0.17 | 0.26 | 0.49 | 0.26 | 0.22 | 0.51 | 0.19 | 0.30 |
| | ASVD | 51 | 70 | 41 | 0.18 | 0.43 | 0.53 | 0.37 | 0.25 | 0.65 | 0.21 | 0.38 |
| | SVD-LLM | 9.56 | 29.44 | 25.11 | 0.20 | 0.48 | 0.59 | 0.40 | 0.26 | 0.65 | 0.22 | 0.40 |
| | Basis Sharing | 9.25 | 29.12 | 22.46 | 0.27 | 0.63 | 0.63 | 0.40 | 0.30 | 0.68 | 0.24 | 0.45 |
| 40% | SVD | 52489 | 59977 | 47774 | 0.15 | 0.26 | 0.52 | 0.26 | 0.22 | 0.53 | 0.20 | 0.30 |
| | FWSVD | 18156 | 20990 | 12847 | 0.16 | 0.26 | 0.51 | 0.26 | 0.22 | 0.53 | 0.21 | 0.30 |
| | ASVD | 1407 | 3292 | 1109 | 0.13 | 0.28 | 0.48 | 0.26 | 0.22 | 0.55 | 0.19 | 0.30 |
| | SVD-LLM | 13.11 | 63.75 | 49.83 | 0.19 | 0.42 | 0.58 | 0.33 | 0.25 | 0.60 | 0.21 | 0.37 |
| | Basis Sharing | 12.39 | 55.78 | 41.28 | 0.22 | 0.52 | 0.61 | 0.35 | 0.27 | 0.62 | 0.23 | 0.40 |
| 50% | SVD | 131715 | 87227 | 79815 | 0.16 | 0.26 | 0.50 | 0.26 | 0.23 | 0.52 | 0.19 | 0.30 |
| | FWSVD | 24391 | 28321 | 23104 | 0.12 | 0.26 | 0.50 | 0.26 | 0.23 | 0.53 | 0.20 | 0.30 |
| | ASVD | 15358 | 47690 | 27925 | 0.12 | 0.26 | 0.51 | 0.26 | 0.22 | 0.52 | 0.19 | 0.30 |
| | SVD-LLM | 23.97 | 150.58 | 118.57 | 0.16 | 0.33 | 0.54 | 0.29 | 0.23 | 0.56 | 0.21 | 0.33 |
| | Basis Sharing | 19.99 | 126.35 | 88.44 | 0.18 | 0.42 | 0.57 | 0.31 | 0.23 | 0.58 | 0.22 | 0.36 |

Table 2: PPL(↓) and Zero-shot(↑) performance of LLaMA2-7B with Basis Sharing under 20% to 50% compression ratios on three language modeling datasets and seven common sense reasoning datasets. The $S$ of all language modeling tasks is evaluated with WikiText-2. For reasoning tasks, the $S$ of the results outside the bracket is evaluated with WikiText-2, while inside is evaluated with Alpaca.

| RATIO | WikiText-2↓ | PTB↓ | C4↓ | Openb. | ARC_e | WinoG. | HellaS. | ARC_c | PIQA | MathQA | Average↑ |
|---|---|---|---|---|---|---|---|---|---|---|---|
| 0% | 5.47 | 7.29 | 7.29 | 0.31 | 0.76 | 0.69 | 0.57 | 0.43 | 0.78 | 0.28 | 0.55 |
| 20% | 7.77 | 60.00 | 15.30 | 0.27 (0.28) | 0.66 (0.70) | 0.63 (0.63) | 0.43 (0.46) | 0.33 (0.35) | 0.70 (0.74) | 0.25 (0.25) | 0.47 (0.49) |
| 30% | 9.69 | 97.40 | 23.86 | 0.26 (0.27) | 0.58 (0.65) | 0.62 (0.62) | 0.38 (0.41) | 0.27 (0.32) | 0.66 (0.70) | 0.23 (0.24) | 0.43 (0.46) |
| 40% | 13.62 | 195.95 | 43.89 | 0.19 (0.21) | 0.48 (0.57) | 0.58 (0.57) | 0.33 (0.36) | 0.22 (0.27) | 0.61 (0.66) | 0.23 (0.23) | 0.38 (0.41) |
| 50% | 21.3 | 509.30 | 98.92 | 0.15 (0.17) | 0.36 (0.47) | 0.55 (0.53) | 0.29 (0.31) | 0.20 (0.25) | 0.56 (0.60) | 0.23 (0.22) | 0.33 (0.36) |

original values. This input deviation affects the evaluations of $S$ with $S(S)^T = cholesky((X)^T X)$. To incorporate this input deviation, we update the weights in the next layers for basis sharing with such deviated inputs, similar to that in SVD-LLM.

## 4.2 RESULTS

We evaluate the performance of the proposed cross-layer parameter sharing from four aspects: (a) Performance on generation and reasoning tasks and comparison with state of the art in zero-shot setting. (b) LLM Performance on different LLMs in zero-shot setting. (c) Performance on LLMs with various scales in zero-shot setting. (d) LLM performance with LoRA (Hu et al., 2021) fine-tuninng. (e) Comparison with training from scratch for weight sharing across layers.

**Performance on Generation & Reasoning Tasks**  We demonstrate the performance of LLaMA-7B and LLaMA2-7B on ten datasets under different compression ratios from 20% to 50%. In evaluating the LLM performance, we group two consecutive layers in the order from the first layer to the last layer to share a basis matrix, while Basis Sharing with more than two layers will be discussed later. Table 10 shows the results of LLaMA-7B. The first three datasets are for text generation tasks and the rest seven datasets are for reasoning tasks. For text generation tasks evaluated by perplexity (PPL), Basis Sharing consistently achieves the lowest PPL among compared with the state-of-the-art methods across all compression ratios and tasks. In reasoning tasks, Basis Sharing achieves an average accuracy at least 3% higher than the state-of-the-art methods. As the compression ratio increases, model performance consistently declines across all the methods due to the incurred larger compression errors. In short, Basis Sharing outperforms SVD-LLM due to smaller compression errors as discussed in Section 3.

Table 3: PPL (↓) of three different LLMs – OPT-6.7B, LLaMA 2-7B, and Mistral-7B – under 20% compression ratio on WikiText-2.

| METHOD | OPT-6.7B | LLaMA 2-7B | Mistral-7B |
|---|---|---|---|
| SVD | 66275 | 18192 | 159627 |
| FWSVD | 14559 | 2360 | 6357 |
| ASVD | 82 | 10.10 | 13.72 |
| SVD-LLM | 16.04 | 8.5 | 10.21 |
| Basis Sharing | **11.79** | **7.70** | **7.57** |

Table 4: PPL (↓) of LLaMA-7B, 13B, 30B under 20% compression ratio on WikiText-2. OOM means out of memory error during the model compression.

| METHOD | LLaMA-7B | LLaMA-13B | LLaMA-30B |
|---|---|---|---|
| SVD | 20061 | 946.31 | 54.11 |
| FWSVD | 1630 | OOM | OOM |
| ASVD | 11.14 | 6.74 | 22.71 |
| SVD-LLM | 7.94 | 6.61 | 5.63 |
| Basis Sharing | **7.74** | **6.47** | **5.47** |

Table 2 presents the basis sharing results of LLaMA2-7B. For the common reasoning tasks, $S$ are evaluated with both WikiText-2 and Alpaca (Taori et al., 2023) to demonstrate the performance difference. The result outside the bracket is based on the evaluation of $S$ with WikiText-2, while the result within the bracket is based on the evaluation of $S$ from Alpaca. It can be seen that LLaMA2-7B is more sensitive to parameter compression, especially on the PTB task. When the compression ratio reaches to 50%, the PPL of LLaMA2-7B is four times of the PPL of LLaMA-7B, while the performance on the remaining tasks are still comparable.

According to Table 2, the input dataset from which $S$ is derived plays a crucial role in determining performance on common reasoning tasks in zero-shot settings. Generally, the model where $S$ is evaluated with Alpaca achieves better accuracy than the model where $S$ is evaluated with WikiText-2, especially on ARC_e under 50% compression ratio. The accuracy difference can reach 11%. However, on WinoG. the difference is not obvious, the model where $S$ is evaluated with WikiText-2 achieves even higher accuracy under 40% and 50% compression ratios.

**Performance on Different LLMs**    To evaluate the generalization of Basis Sharing across multiple LLMs, we evaluate its PPL on three distinct models from three LLM families: OPT-6.7B (from the OPT family), LLaMA 2-7B (from the LLaMA family), and Mistral-7B (from the Mistral family). This comparison is conducted under a 20% compression ratio using the WikiText-2 dataset without any fine-tuning. It can be seen from Table 3, Basis Sharing consistently achieves the lowest PPL. Especially for OPT-6.7B and Mistral-7B, Basis Sharing achieves a PPL reduction up to 25% compared with SVD-LLM.

**Performance on LLMs with Various Scales**    Basis Sharing can be applied to LLMs with large scales. To demonstrate this, we apply Basis Sharing on LLaMA with three different scales under 20% compression ratio, namely LLaMA-7B, LLaMA-13B and LLaMA-30B against the state-of-the-art methods. The result is shown in Table 4. According to this table, Basis Sharing achieves the best performance across all the scales. Since gradient needs to be computed with FWSVD, out of memory error occurs on an A100 GPU. In contrast, Basis Sharing can still be realized with an A100 GPU.

**Performance with LoRA Fine-Tuning**    LoRA (Hu et al., 2021) is one of the most promising fine-tuning techniques to recover performance/accuracy. LoRA can also be applied to Basis Sharing to recover performance/accuracy. We used *lora_r* = 8, *lora_alpha* = 32, and *learning_rate* = 1e-4, and used defaults for all other hyperparameters in the Hugging Face PEFT. Each model is fine tuned with WikiText-2 training dataset for two epochs.

Figure 5 shows the result after applying LoRA on LLaMA-7B with WikiText-2. It can be seen from the figure that all compression methods achieve similar PPL under 20% compression ratio, and PPL difference increases as the compression ratio goes up. Basis Sharing achieves the lowest PPL when the compression ratio reaches 50%.

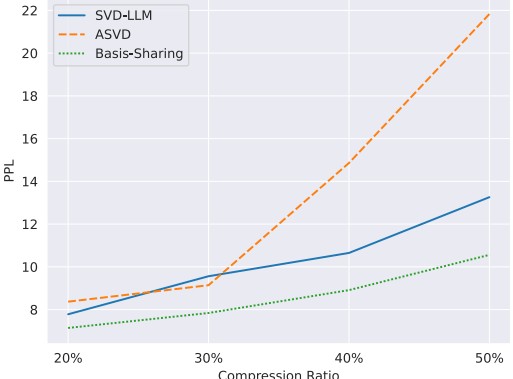

Figure 5: LoRA fine-tuning results of LLaMA-7B under 20% compression ratio with different compression methods.

Table 6: Impact of grouping different numbers of layers on LLaMA-7B under compression ratios from 20% to 50%.

| # LAYERS | 20% | 30% | 40% | 50% |
|---|---|---|---|---|
| 1 | 7.94 | 9.56 | 13.11 | 23.97 |
| 2 | 7.74 | 9.25 | **12.39** | **19.99** |
| 3 | 7.72 | 9.27 | 12.60 | 20.06 |
| 4 | 7.65 | **9.18** | 12.58 | 20.86 |
| 5 | **7.62** | 9.19 | 12.81 | 24.45 |
| 6 | 7.64 | 9.20 | 14.13 | 25.40 |
| 7 | 7.67 | 9.24 | 14.64 | 27.30 |
| 8 | 7.75 | 9.49 | 14.60 | 27.92 |
| 16 | 7.95 | 10.58 | 19.72 | 49.11 |
| 32 | 7.94 | 9.56 | 30.82 | 85.24 |

Table 7: Impact of grouping different numbers of layers on LLaMA-7B under compression ratios from 20% to 50% after LoRA Fine-Tuning.

| # LAYERS | 20% | 30% | 40% | 50% |
|---|---|---|---|---|
| 1 | 7.78 | 9.56 | 10.65 | 13.26 |
| 2 | 7.14 | 7.84 | 8.91 | 10.56 |
| 3 | 7.00 | 7.81 | 9.04 | 10.35 |
| 4 | 7.07 | 7.86 | 9.02 | 10.36 |
| 5 | 6.98 | 8.05 | 9.23 | 10.14 |
| 6 | 6.88 | 8.03 | 9.06 | 10.32 |
| 7 | 6.75 | 7.57 | 9.08 | 10.76 |
| 8 | 6.89 | 7.68 | 9.14 | 10.32 |
| 16 | 7.02 | 7.82 | 9.27 | 11.20 |
| 32 | 6.97 | 8.25 | 9.37 | 11.64 |

Table 5: GPT2 20% compression ratio compared with Dynamic Tying.

| METHOD | # Parm. | Time | PPL |
|---|---|---|---|
| Dynamic Tying | 264M (GPT2-XL) | 13.75h | 49.37 |
| Basis Sharing | 94M (GPT2) | 26.47s | 43.15 |

**Comparison with Training from Scratch** Table 5 compares Basis Sharing with Dynamic Tying(Hay & Wolf, 2024), where parameter sharing is realized by training from scratch. Instead of training from scratch, Basis Sharing leverages pretrained models that have been trained on large datasets and trained with more computational resources. As a result, Basis Sharing achieves fewer parameters, faster compression, and better PPL on WikiText-2 compared to Dynamic Tying.

## 4.3 IMPACT OF LAYER SELECTION OF BASIS SHARING ON LLM PERFORMANCE

In section 3, we analyzed the change of Frobenius loss when two layers are grouped to share a set of basis vectors. In this section, we will demonstrate how grouping more than two consecutive layers affects the LLM performance.

**Impact on LLM Performance in Zero-Shot Setting** We grouped different numbers of consecutive layers to examine the impact of the number grouped layers on the LLM performance without any fine-tuning. Table 6 shows the result. The number in the first column indicates the number of consecutive layers sharing a common basis matrix. For example, 4 means that every four consecutive layers share a basis matrix in the order from the first layer to the last layer. Compared with no basis sharing in SVD-LLM (# LAYERS = 1) under 20% compression ratio, Basis Sharing achieves a similar performance. Grouping four or five layers to share a basis matrix is more reasonable when compression ratio is lower than 30%, since they have the lowest PPL. Two layers sharing a basis matrix is a good choice when the compression ratio is larger than 30%.

**Impact on LLM Performance with LoRA Fine-Tuning** We also examined the impact of grouping different number of layers on LLM performance after LoRA Fine-Tuning. Table 7 shows the result. According to this table, the performance of LLM can be enhanced compared with that without fine-tuning. In addition, this table also shows that after LoRA fine-tuning, grouping layers in LLaMA-7B for Basis Sharing can achieve better performance than that without basis sharing in SVD-LLM (# LAYERS = 1). Even when the number of grouped layer is 32, the performance of Basis Sharing is still better than that without basis sharing in SVD-LLM (# LAYERS = 1).

**Impact on LLM Peformance with Full Parameter Fine-Tuning** To examine the full potential of the Basis Sharing, we also conducted the full parameter fine-tuning to examine the impact of grouping different numbers of layers on LLM performance. Due to the high computational cost, we only fine tuned the LLaMA-7B on grouping 2, 4, 8, 16, 32 layers, respectively. The differences from LoRA fine-tuning are that we use here *learning_rate* = 2e-6 and two A100 GPUs. The results of full parameter fine-tuning can be found in Table 8. It can be seen that the performance with full parameter fine-tuning is only a little bit better than the performance with LoRA fine-tuning. The reason could be

that WikiText-2 is relatively a small dataset to fine-tune the large model. Directly using this dataset to fine-tune could easily lead to overfitting. Therefore, we reduce the *learning_rate* from 1e-4 to 2e-6.

### 4.4 PERFORMANCE ON REAL HARDWARE

Basis Sharing not only reduces the memory required for storing parameters, but also enhances inference efficiency on real hardware. To demonstrate this advantage, we compared the performance of LLaMA-7B with and without Basis Sharing on a single A100 GPU, using a batch size of 512 and a sequence length of 32 to generate one token for each batch. With this setting, throughput was evaluated as the total number of tokens that can be processed by the model per second.

Table 8: Result of full parameter fine-tuning by grouping different numbers of layers.

| # LAYERS | 20% | 30% | 40% | 50% |
|----------|------|------|------|-------|
| 2 | 6.57 | 7.41 | 8.29 | 9.71 |
| 4 | 6.64 | 7.39 | 8.41 | 9.91 |
| 8 | 6.63 | 7.46 | 8.54 | 10.23 |
| 16 | 6.66 | 7.66 | 9.04 | 10.48 |
| 32 | 6.67 | 7.90 | 9.24 | 10.94 |

## 5 CONCLUSION

Figure 6 shows the throughput result. It can be seen that as the compression ratio increases, the throughput of model with Basis Sharing also increases. Under 50% compression ratio, the throughput of Basis Sharing is 1.57 times of the dense model. In this paper, we explore parameter sharing across different layers with SVD to achieve effective compression for LLMs. Specifically, weight matrices in different layers are decomposed and represented as a linear combination of a set of shared basis vectors and unique coefficients. The types of weight matrices and the layer selection for Basis Sharing are examined when compressing LLMs to maintain the performance. Comprehensive experiments demonstrate that Basis Sharing outperforms state-of-the-art SVD-based compression approaches, especially under large compression ratios.

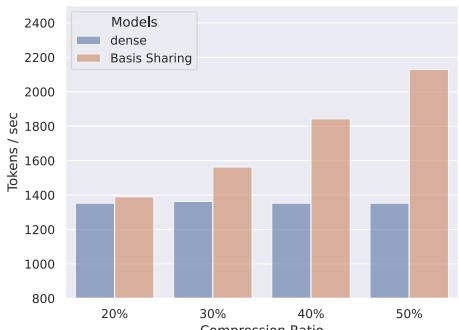

Figure 6: Throughput of dense LLaMA-7B model and the compressed model with Basis Sharing under compression ratios from 20% to 50%.

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

# A APPENDIX

## A.1 FINAL STRUCTURE OF TWO LAYERS IN LLaMA-7B WITH BASIS SHARING

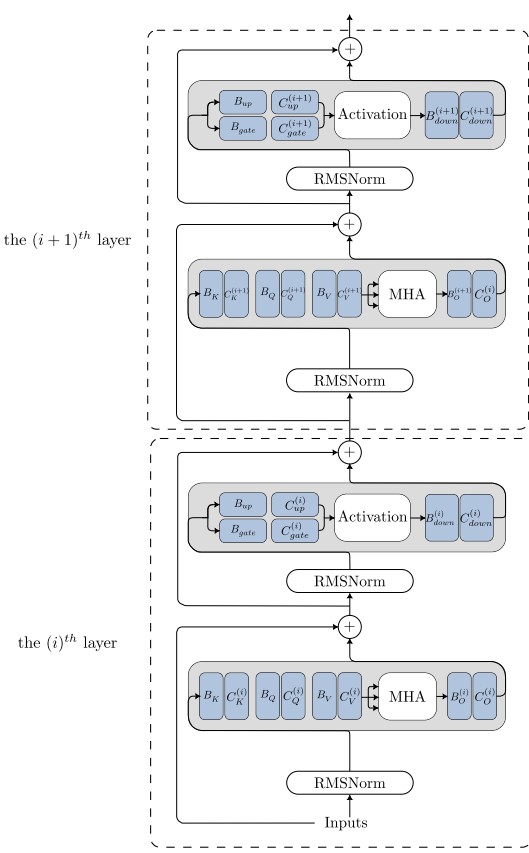

Figure 7: The final structure of two layers in LLaMA-7B with Basis Sharing. MHA represents multi-head attention. RMSNorm represents root mean square of layer normalization.

## A.2 RELATION BETWEEN COMPRESSION RATIO AND NUMBER OF BASIS VECTORS

For a given compression ratio, the derivation of the number of basis vectors $k$ is explained as follows. Consider compressing $W_K$ weight matrices in consecutive $n$ layers to x% of their original sizes. Assume each $W_K$ matrix have $d_1$ rows and $d_2$ columns. The number of basis vectors $k$ can be calculated as follows:

$$d_1 k + k d_2 n = d_1 d_2 n \times x\% \Rightarrow k = \frac{d_1 d_2 n \times x\%}{(d_1 + d_2 n)}$$

where $d_1 d_2 n$ is the number of parameters of $W_K$ weight matrices in $n$ layers before compression and $d_1 k + k d_2 n$ is the number of parameters after sharing basis vectors for weight matrices in consecutive $n$ layers.

To compare with traditional SVD methods, the same compression ratios were used to evaluate the rank of the weight matrix in each layer individually. Consider compressing $W_K$ weight matrix to x% of its original size. Assume this matrix have $d_1$ rows and $d_2$ columns. The rank of this matrix $k$ can be calculated as follows:

$$d_1 k + k d_2 = d_1 d_2 \times x\% \Rightarrow k = \frac{d_1 d_2 \times x\%}{d_1 + d_2}$$

Under the same compression ratio (1-x%), basis sharing can lead to a larger $k$ compared with that with traditional SVD-LLM, so that the performance of LLMs can be enhanced.

A.3   ANALYSIS OF MATHEMATICAL PROPERTIES OF MATRICES SHARED ACROSS LAYERS

Suppose $A = SW$ is a matrix of the $i^{th}$ layer, which has $d_1$ rows and $d_2$ columns. $S$ is the scaling matrix imposed on original weight matrix to incorporate the impact of input data. Assume that we want to apply Basis Sharing on $n$ such matrices in $n$ layers, where $n >= 2$. $B$ is the horizontal concatenation of such $n$ matrices, which has $d_1$ rows and $nd_2$ columns. We analyzed the Frobenius loss $F\_loss$ incurred by compression without and with basis sharing as follows. In the following equations, $x\%$ represents to compress the matrix to $x\%$ of its original size. The maximum value of $x$ is 100. $k_{svd}$ and $k_{share}$ represent the number of top singular values after SVD is applied in each layer and the number of basis vectors after SVD is applied in the concatenated matrices of $n$ layers, respectively. $F\_loss_{svd}$ and $F\_loss_{share}$ represent the Frobenius loss without and with basis sharing, respectively. $\sigma_i$ is the $i$th removed singular value after SVD decomposition. $\sigma_{svd}$ is the average singular value after applying SVD decomposition on $A$. $\sigma_{share}$ is the average singular value after applying SVD decomposition on $B$.

**Case 1**: $d_1 \leq d_2$, rank(A)=rank(B)=$d_1$

$$k_{svd} = \frac{d_1 d_2}{d_1 + d_2} x\% = \frac{x\%}{\frac{1}{d_1} + \frac{1}{d_2}} \geq \frac{1}{2} d_1 x\%$$

$$F\_loss_{svd} \leq \sum_{i=k_{svd}}^{d_1} \sigma_i \approx (d_1 - \frac{1}{2} d_1 x\%) \sigma_{svd}$$

$$k_{share} = \frac{n d_1 d_2}{d_1 + nd_2} x\% = \frac{x\%}{\frac{1}{d_1} + \frac{1}{nd_2}} \geq \frac{n}{n+1} d_1 x\%$$

$$F\_loss_{share} \leq \sum_{i=k_share}^{d_1} \sigma_i \approx (d_1 - \frac{n}{n+1} d_1 x\%) \sigma_{share}$$

In case that $\sigma_{svd} = \sigma_{share} = \sigma$, we can derive the following relationship:

$$max(nF\_loss_{svd}) - max(F\_loss_{share}) = (n-1)(d_1 - \frac{n}{2(n+1)} d_1 x\%) \sigma > 0$$

In this case, we have $max(nF\_loss_{svd}) > max(F\_loss_{share})$, which indicates basis sharing across $n$ layers can reduce the upper bound of the Frobenius loss and potentially reduce the the Frobenius loss. In our work $W_K$, $W_Q$, $W_V$, $W_{up}$ and $W_{gate}$ in LLaMA-7B have such mathematical properties and thus can benefit from this basis sharing. However, for $W_O$, the assumption of $\sigma_{svd} = \sigma_{share} = \sigma$ does not hold and $\sigma_{share}$ is much larger than $\sigma_{svd}$, so that the Frobenius loss with sharing is larger than that without sharing. Accordingly, such a matrix can not take advantage of basis sharing across layers.

**Case 2**: $d_1 \geq nd_2$, rank(A)=$d_2$, rank(B)=$nd_2$

$$k_{svd} = \frac{x\%}{\frac{1}{d_1} + \frac{1}{d_2}} \geq \frac{n}{n+1} d_2 x\%$$

$$F\_loss_{svd} \leq \sum_{i=k_{svd}}^{d_2} \sigma_i \approx (d_2 - \frac{n}{n+1} d_2 x\%) \sigma_{svd}$$

$$k_{share} = \frac{x\%}{\frac{1}{d_1} + \frac{1}{nd_2}} \geq \frac{n}{2} d_2 x\%$$

$$F\_loss_{share} \leq \sum_{i=k_{share}}^{nd_2} \sigma_i \approx (nd_2 - \frac{n}{2} d_2 x\%) \sigma_{share}$$

In case that $\sigma_{svd} = \sigma_{share} = \sigma$, we can derive the following relationship:

$$max(nF\_loss_{svd}) - max(F\_loss_{share}) = (\frac{n}{2} d_2 x\% - \frac{n^2}{n+1} d_2 x\%) \sigma < 0$$

In this case, we have $max(nF\_loss_{svd}) < max(F\_loss_{share})$, which indicates basis sharing can increase the upper bound of the Frobenius loss and potentially increase the Frobenius loss.

In our work, $W_{down}$ in LLaMA-7B has such mathematical properties when $n = 2$ and thus can not benefit from this basis sharing.

**Case 3**: $d_2 < d_1 < nd_2$, rank(A)=$d_2$, rank(B)=$d_1$

$$k_{svd} = \frac{x\%}{\frac{1}{d_1} + \frac{1}{d_2}} > \frac{1}{n+1}d_1 x\%$$

$$F\_loss_{svd} = \sum_{i=k_{svd}}^{d_2} \sigma_i \approx (d_2 - \frac{1}{n+1}d_1 x\%)\sigma_{svd}$$

$$k_{share} = \frac{x\%}{\frac{1}{d_1} + \frac{1}{nd_2}} > \frac{1}{2}d_1 x\%$$

$$F\_loss_{share} < \sum_{i=k_{share}}^{d_1} \sigma_i \approx (d_1 - \frac{1}{2}d_1 x\%)\sigma_{share}$$

In case that $\sigma_{svd} = \sigma_{share} = \sigma$, we can derive the following relationship:

$$max(nF\_loss_{svd}) - max(F\_loss_{share}) = (nd_2 - d_1 + \frac{1-n}{2(n+1)}d_1 x\%)\sigma$$

$$-\frac{n-1}{2(n+1)}d_1 x\%\sigma < (nd_2 - d_1 + \frac{1-n}{2(n+1)}d_1 x\%)\sigma < (nd_2 - d_1)\sigma$$

In this case, whether basis sharing across layers has potential to reduce the Frobenius loss cannot be determined. In our work, $W_{down}$ in LLaMA-7B has such mathematical properties when $n >= 3$ and we decide not to share basis for $W_{down}$ across layers in LLaMA-7B.

**Future work** To reduce the Frobenius loss after basis sharing, we will explore the potential of vertically concatenating $n$ matrices across layers. The vertically concatenated $B$ has $nd_1$ rows and $d_2$ columns. In this case, there is still potential to reduce the Frobenius loss as follows.

For such a matrix $d_2 < d_1$ and rank(A)=rank(B)=$d_2$

$$k_{svd} = \frac{x\%}{\frac{1}{d_1} + \frac{1}{d_2}} > \frac{1}{2}d_2 x\%$$

$$F\_loss_{svd} < \sum_{i=k_{svd}}^{d_2} \sigma_i \approx (d_2 - \frac{1}{2}d_2 x\%)\sigma_{svd}$$

$$k_{share} = \frac{x\%}{\frac{1}{d_1} + \frac{1}{nd_2}} > \frac{n}{n+1}d_2 x\%$$

$$F\_loss_{share} < \sum_{i=k_{share}}^{d_2} \sigma_i \approx (d_2 - \frac{n}{n+1}d_2 x\%)\sigma_{share}$$

In case that $\sigma_{svd} = \sigma_{share} = \sigma$, we can derive the following relationship:

$$max(nF\_loss_{svd}) - max(F\_loss_{share}) = (n-1)(d_2 - \frac{n}{2(n+1)}d_2 x\%) > 0$$

In this case, the upper bound of Frobenius loss with basis sharing can be reduced. For weight matrix such as $W_{down}$, we will concatenate such matrices across $n$ layers vertically and decompose the concatenated matrix to obtain their basis vectors.

However, the computation of scaling matrix $S$ to consider the impact of activations becomes more time-consuming due to the increasing number of rows. We will address this challenge in our follow-up work.

## A.4 EVALUATING ZERO-SHOT COMMON-SENSE REASONING TASKS AFTER LORA FINE-TUNING

In this section, we will show that LoRA fine-tuning can also enhance the accuracy of zero-shot common-sense reasoning tasks.

| Ratio | Openb. | ARC_e | WinoG. | HellaS. | ARC_c | PIQA | MathQA | Avg |
|---|---|---|---|---|---|---|---|---|
| **20%** | 0.28(0.28) | 0.67(0.67) | 0.66(0.66) | 0.49(0.46) | 0.35(0.36) | 0.72(0.71) | 0.25(0.25) | 0.49(0.48) |
| **30%** | 0.28(0.27) | 0.63(0.63) | 0.64(0.63) | 0.45(0.40) | 0.32(0.30) | 0.7(0.68) | 0.25(0.24) | 0.47(0.45) |
| **40%** | 0.24(0.22) | 0.54(0.52) | 0.60(0.61) | 0.40(0.35) | 0.29(0.27) | 0.66(0.62) | 0.24(0.23) | 0.42(0.40) |
| **50%** | 0.22(0.18) | 0.49(0.42) | 0.59(0.57) | 0.36(0.31) | 0.24(0.23) | 0.62(0.58) | 0.22(0.22) | 0.39(0.36) |

Table 9: The performance on zero-shot common-sense reasoning tasks using LLaMA-7B compressed with Basis Sharing, with and without LoRA fine-tuning. The number in the bracket is without LoRA fine-tuning.

## A.5 PERFORMANCE OF LLAMA3.2-3B WITH BASIS SHARING

Table 10: Zero-shot performance of LLaMA-3.2B compressed using Basis Sharing and baselines under 20% to 50% compression ratios on WikiText-2 (measured by perplexity (↓)) and seven common-sense reasoning datasets (measured by both individual and average accuracy (↑)).

| RATIO | METHOD | WikiText-2 ↓ | Openb. | ARC_e | WinoG. | HellaS. | ARC_c | PIQA | MathQA | **Average** ↑ |
|---|---|---|---|---|---|---|---|---|---|---|
| 0% | Original | 7.84 | 0.31 | 0.75 | 0.70 | 0.55 | 0.42 | 0.77 | 0.35 | 0.55 |
| 20% | SVD-LLM | 38.39 | 0.19 | 0.53 | 0.57 | 0.33 | 0.24 | 0.63 | 0.24 | 0.39 |
| | Basis Sharing | 22.48 | 0.20 | 0.54 | 0.58 | 0.35 | 0.25 | 0.65 | 0.25 | 0.40 |
| 30% | SVD-LLM | 44.22 | 0.14 | 0.41 | 0.54 | 0.30 | 0.19 | 0.59 | 0.23 | 0.34 |
| | Basis Sharing | 27.41 | 0.15 | 0.44 | 0.56 | 0.30 | 0.20 | 0.59 | 0.23 | 0.35 |
| 40% | SVD-LLM | 65.09 | 0.12 | 0.34 | 0.54 | 0.28 | 0.18 | 0.55 | 0.23 | 0.32 |
| | Basis Sharing | 59.95 | 0.14 | 0.34 | 0.54 | 0.28 | 0.19 | 0.56 | 0.23 | 0.33 |
| 50% | SVD-LLM | 106.42 | 0.12 | 0.31 | 0.51 | 0.27 | 0.18 | 0.54 | 0.22 | 0.30 |
| | Basis Sharing | 104.69 | 0.12 | 0.31 | 0.49 | 0.27 | 0.19 | 0.54 | 0.23 | 0.30 |

## A.6 COMPRESSION GAINS

To demonstrate the compression gains through layer sharing, we did two further experiments. In the first experiment, we used SVD to decompose weight matrices in each layer of LLaMA-7B and compressed matrices with 20% compression ratio. Under this compression ratio, we evaluated how many top $k$ singular values were kept in the $\Sigma$ after SVD decomposition. When basis sharing is applied to group every 2, 4, 8, 16 and 32 consecutive layers, the same value of $k$ was used as the number of basis vectors to evaluate the model performance after basis sharing. The results are shown in the following left table. According to this table, with more layers shared, the compression ratio increases while the performance degrades without LoRA fine-tuning. However, the performance can be enhanced significantly after LoRA fine-tuning.

In the second experiment, 30% compression ratio was used to compress weight matrices in each layer to evaluate the number of top singular values $k$ kept in the $\Sigma$ after SVD decomposition. Afterwards, this number was used to evaluate the performance of basis sharing, the result of which is shown in the following right table. Similarly, compression ratios increase when basis sharing is enabled. The performance of basis sharing can still be enhanced by LoRA fine-tuning.

Table 11: Compression gain with basis sharing, start from 20% compression ratio. $\#Layers$ is the number of shared layers. $PPL'$ is the PPL after LoRA fine-tuning.

| #Layers | Comp. Ratio | PPL | $PPL'$ |
|---|---|---|---|
| 1 | 20% | 7.94 | 7.78 |
| 2 | 29% | 8.94 | 7.52 |
| 4 | 34% | 10.1 | 8.15 |
| 8 | 36% | 11.99 | 8.27 |
| 16 | 37% | 20.99 | 9.16 |
| 32 | 38% | 35.48 | 9.45 |

Table 12: Compression gain with basis sharing, start from 30% compression ratio. $\#Layers$ is the number of shared layers. $PPL'$ is the PPL after LoRA fine-tuning.

| #Layers | Comp. Ratio | PPL | $PPL'$ |
|---|---|---|---|
| 1 | 30% | 9.56 | 9.14 |
| 2 | 37% | 11.32 | 8.74 |
| 4 | 42% | 13.56 | 9.12 |
| 8 | 43% | 19.72 | 9.48 |
| 16 | 44% | 35 | 10.57 |
| 32 | 45% | 93.85 | 11.00 |

## A.7 GENERATED TEXT WITH COMPRESSED LLM

| RATIO | BASIS SHARING |
|---|---|
| Original | **What is the universe?** The universe is a vast collection of galaxies and stars. The Sun, Earth, Moon are all part of this Universe which includes everything that can be seen with our naked eyes or telescopes such as... |
| 20% | **What is the universe?** The universe is a huge collection of interstellar objects. The Sun is one such object and, in fact we are located within this vast system known as our home star system (the solar system)... |
| 30% | **What is the universe?** The universe is a gigantic system of stars held together by gravity, which binds them to each other. The Sun has been at its present distance from Earth since it formed over 4 billion years ago... |
| 40% | **What is the universe?** The universe is a giant star system that contains many stars and planet systems. The Milky Way, the galaxy containing our solar system, has two main components: the inner part of the system composed of small gas... |
| 50% | **What is the universe?** The universe is a large collection of objects, stars. These stars are arranged in layers and form different stellar classes . The outer solar regions have many denser stars called main sequences with massive hydrogen masses, which... |

Table 13: An example of contents generated by the compressed LLaMA-7B with Basis Sharing under different compression ratios. The input is marked in bold and the normal texts are the generated sentences.

## A.8 SHARE ERROR HEAT MAP

The Frobenius loss inccured by basis sharing for $\boldsymbol{W}_Q$ , $\boldsymbol{W}_V$, $\boldsymbol{W}_{Up}$ and $\boldsymbol{W}_{Gate}$.

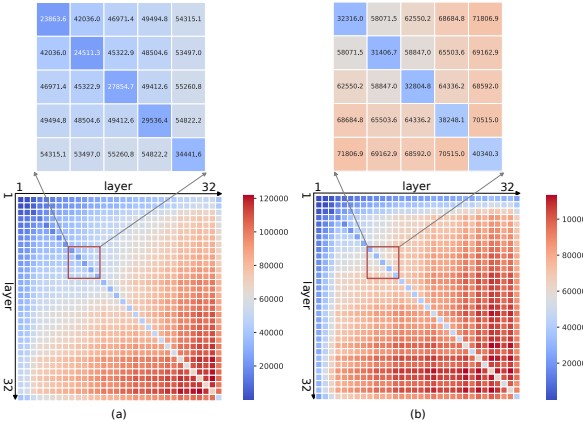

Figure 8: Frobenius loss incurred by basis sharing across any two layers. The number/color in a block represents the resulting Frobenius loss if a basis matrix is shared by two layers and the numbers in the diagonal direction are obtained by applying SVD to the scaled weight matrix of a layer directly. (a) Frobenius loss incurred by basis sharing across two layers for $\boldsymbol{W}_Q$ in LLaMA2-7B. (b) Frobenius loss incurred by basis sharing across two layers for $\boldsymbol{W}_V$ in LLaMA2-7B.

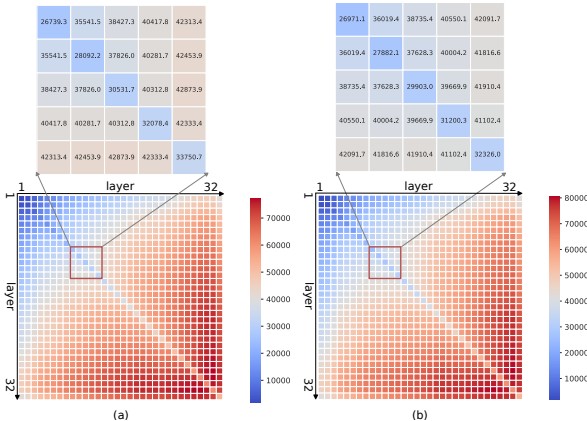

Figure 9: Frobenius loss incurred by basis sharing across any two layers. The number/color in a block represents the resulting Frobenius loss if a basis matrix is shared by two layers and the numbers in the diagonal direction are obtained by applying SVD to the scaled weight matrix of a layer directly. (a) Frobenius loss incurred by basis sharing across two layers for $W_{Up}$ in LLaMA2-7B. (b) Frobenius loss incurred by basis sharing across two layers for $W_{Gate}$ in LLaMA2-7B.

