# OpenReview forum: "Basis Sharing: Cross-Layer Parameter Sharing for Large Language Model Compression"
_ICLR.cc/2025/Conference — ICLR 2025 Poster_

### Official Review · Reviewer_jR33 · 2024-10-28

**Soundness:** 3
**Presentation:** 3
**Contribution:** 3
**Rating:** 5
**Confidence:** 3

**Summary:**

The paper proposes a novel layer-sharing SVD compression strategy for LLMs, building upon previous SVD compression methods. The authors suggest that by sharing the left singular matrix across different layers, the model can achieve better performance. Experiments demonstrate that this method outperforms existing activation-aware SVD compression techniques.

**Strengths:**

1. The proposed idea of sharing the left singular matrix between layers is insightful. This approach significantly improves the compression ratio of the model.
2. The finding that model performance can be improved through layer sharing is intriguing.
3. The experiments are comprehensive.

**Weaknesses:**

1. While the paper claims that layer sharing can improve performance, it does not explain why. The paper lacks an explanation of the properties of the weight matrices that enable this improvement, making the conclusion somewhat abrupt and counterintuitive.
2. Sharing left singular vectors may not be applicable to all matrices in the model, which suggests that this is not a general approach. The paper does not provide in-depth analysis of this phenomenon.
3. Using layer sharing implies that the truncated ranks of both layers are identical, which could limit the model's performance.

**Questions:**

1. In the experiments section, the paper provides results for different compression ratios. However, I did not see an explanation in the paper on how the specific compression ratios were determined. Could you provide further clarification? And how did the authors determine the cutoff values ​​for different layers?
2. Could the authors analyze the properties of matrices that can be shared across layers, particularly their mathematical properties, and explain why performance is improved? This result seems should be sensitive to the training data and is counterintuitive, as, in theory, not sharing should provide a higher performance ceiling compared to sharing.
3. Could the authors provide the actual number of layers that can be shared in the model and the compression gains achieved through layer sharing?

---

> ### Author Response · Authors · 2024-11-26
>
> Hi Reviewer jR33,
>
> Many thanks for pointing out the lack of clarity in the evaluation of the number of basis vectors and the analysis of mathematical properties of matrices that can be shared across layers. We have explained them in details as follows and added the corresponding evaluation and analysis in the Appendix of the paper. We hope the explanation could clear your doubts. Thanks again for your time and insightful feedback.
>
> **Answer to Question 1: Evaluation of different compression ratios in different layers**
>
> Thanks for pointing out this lack of clarity. In experiments, we first set a given compression ratio, e.g., 20\%, and used this compression ratio to derive the number of basis vectors $k$ for grouping consecutive $n$ layers. To demonstrate the advantage of Basis Sharing, different compression ratios were used to compare the performance of the proposed method with traditional SVD methods.
>
> For a given compression ratio, the derivation of the number of basis vectors $k$ is explained as follows. Consider compressing $W_K$ weight matrices in consecutive $n$ layers to x\% of their original sizes. Assume each $W_K$ matrix have $d_1$ rows and $d_2$ columns.
> The number of basis vectors $k$ can be calculated as follows:
> $$
>     d_1k + kd_2n = d_1d_2n \times x\\% \Rightarrow k = \frac{d_1d_2n\times x\\%}{(d_1+d_2n)}
> $$
> where $d_1d_2n$ is the number of parameters of $W_K$ weight matrices in $n$ layers before compression and $d_1k + kd_2n$ is the number of parameters after sharing basis vectors for weight matrices in consecutive $n$ layers.
>
> We used this equation to evaluate the number of basis vectors $k$ when grouping every $n$ consecutive layers for basis sharing. This number $k$ is set to be the same for all the group layers. In the future, we will explore non-uniform allocation of $k$ in different grouped layers to further enhance the performance of LLMs.
>
>  To compare with traditional SVD methods, the same compression ratios were used to evaluate the rank of the weight matrix in each layer individually. Consider compressing $W_K$ weight matrix to x\% of its original size. Assume this matrix has $d_1$ rows and $d_2$ columns. The rank of this matrix $k$ can be calculated as follows:
> $$
>     d_1k + kd_2 = d_1d_2 \times x\\% \Rightarrow k = \frac{d_1d_2\times x\\%}{d_1+d_2}
> $$
>
> Under the same compression ratio (1-x\%), basis sharing can lead to a larger $k$ compared with that with traditional SVD-LLM, so that the performance of LLMs can be enhanced. We have added this explanation in the Appendix A.2 on page 13.
>
> **Answer to Question 2: Mathematical properties of matrices that can be shared across layers and the result sensitivity to calibration data.**
>
> Thanks for your question. In short, the weight matrices whose basis sharing across layers leads to a smaller Frobenius loss than that without sharing can be shared. We analyze the properties of matrices that achieve basis sharing across layers in details and included this analysis in Appendix A.3 on page 14-15.
>
> To evaluate the sensitivity of basis sharing to the calibration dataset, we have verified the PPL of LLaMA-7B with different number of batch sizes. The results are shown in the following table. According to this table, after the calibration batch size reaches 128, the PPL is becoming stable. In our experiments, we used the batch size 256 as the calibration batch size.
>
> | **Batch Size** | **16** | **32** | **64** | **128** | **256** | **512** |
> |----------------|---------|---------|---------|----------|----------|----------|
> | **PPL**       | 9.08    | 8.45    | 8.07    | 7.85     | 7.74     | 7.64     |
>
> Table: PPL of LLaMA-7B with different calibration batch sizes.

---

> ### Author Response · Authors · 2024-11-26
>
> **Answer to Question3: Compression gains with increasing number of shared layers**
>
> Thanks for pointing out this lack of clarity. In fact, we have verified the model performance not only by grouping every two consecutive layers but also
> by grouping more than two layers, i.e., 3,4,5,6,7,8,16,32. to explore the potential of Basis Sharing. The results are shown in Table 6 and Table 7 in the paper on page 9. According to the two tables, allowing all the layers in LLaMA-7B to share the same set of basis vectors is possible and the performance can still be maintained after LoRA fine-tuning.
>
> To demonstrate the compression gains through layer sharing, we did two further experiments. In the first experiment, we used SVD to decompose weight matrices in each layer of LLaMA-7B and compressed matrices with 20\% compression ratio. Under this compression ratio, we evaluated how many top $k$ singular values were kept in the $\Sigma$ after SVD decomposition. When basis sharing is applied to group every 2, 4, 8, 16 and 32 consecutive layers, the same value of $k$ was used
> as the number of basis vectors
> to evaluate the model performance after basis sharing. The results are shown in the following left (first) table. According to this table, with more layers shared, the compression ratio increases while the performance degrades without LoRA fine-tuning. However, the performance can be enhanced significantly after LoRA fine-tuning.
>
> In the second experiment, 30\% compression ratio was used to compress weight matrices in each layer to evaluate the number of top singular values $k$ kept in the  $\Sigma$ after SVD decomposition. Afterwards, this number was used to evaluate the performance of basis sharing, the result of which is shown in the following right (second) table. Similarly, compression ratios increase when basis sharing is enabled. The performance of basis sharing can still be enhanced by LoRA fine-tuning.
>
> We have added the two tables in the Appendix A.6 Table 11, 12 of the paper on page 16-17.
>
> | **#Layers** | **Comp. Ratio** | **PPL** | **PPL'** |
> |-------------|-----------------|---------|----------|
> | 1           | 20%            | 7.94    | 7.78     |
> | 2           | 29%            | 8.94    | 7.52     |
> | 4           | 34%            | 10.10    | 8.15     |
> | 8           | 36%            | 11.99   | 8.27     |
> | 16          | 37%            | 20.99   | 9.16     |
> | 32          | 38%            | 35.48   | 9.45     |
>
> Left Table: Compression gain with basis sharing, starting from 20% compression ratio. #Layers is the number of shared layers. PPL' is the PPL after LoRA fine-tuning.
>
> | **#Layers** | **Comp. Ratio** | **PPL** | **PPL'** |
> |-------------|-----------------|---------|----------|
> | 1           | 30%            | 9.56    | 9.14     |
> | 2           | 37%            | 11.32   | 8.74     |
> | 4           | 42%            | 13.56   | 9.12     |
> | 8           | 43%            | 19.72   | 9.48     |
> | 16          | 44%            | 35.00      | 10.57    |
> | 32          | 45%            | 93.85   | 11.00    |
>
> Right Table: Compression gain with basis sharing, starting from 30% compression ratio. #Layers is the number of shared layers. PPL' is the PPL after LoRA fine-tuning.

---

### Official Review · Reviewer_NiaD · 2024-10-31

**Soundness:** 4
**Presentation:** 3
**Contribution:** 3
**Rating:** 8
**Confidence:** 3

**Summary:**

This paper introduces “Basis Sharing,” a method for compressing large language models by sharing basis vectors of SVD across layers, each with unique coefficients.  Basis Sharing significantly reduces model size while maintaining performance. Experiments demonstrate its superiority over traditional SVD-based techniques across various tasks and settings, achieving enhanced efficiency and accuracy at high compression ratios.

**Strengths:**

* This paper analyzes the feasibility of SVD for different modules and outliers in depth, and designs a simpler and more general SVD compression method for LLM: BASIS SHARING.

* The advantages and robustness of the proposed method are fully verified on various tasks, different scales, software and hardware Settings.

* The presentation of the technical discussion is accurate and well-organized.

**Weaknesses:**

* It looks like BASIS SHARING will also have better acceleration in hardware, which I think would be better if compared to other SVD-based methods.

* In this paper, continuous groups are designed based on the interesting insights in 3.2. However, ablation experiments on discontinuous groups are still missing. Is it possible to realize CROSS-LAYER PARAMETER SHARING of functions (such as W_O) that are not suitable for continuous groups through discontinuous groups?

**Questions:**

* Can this method be used in conjunction with other compression methods (e.g. pruning, quantization, etc.) to go one step further?

---

> ### Author Response · Authors · 2024-11-26
>
> Hi Reviewer NiaD,
>
> Thank you very much for your positive and encouraging comments on our work. We will continue to enhance our methods in the follow-up work by combining pruning and quantization as well as exploring the possibility of sharing basis vectors across discontinuous layers. We have addressed your questions as follows.
>
> **Reply to Weakness 1: Hardware performance comparison with other SVD based methods**
>
> Due to time limit, we did two further preliminary experiments to verify the enhancement of hardware performance such as throughput with Basis Sharing. In these experiments, we maintained the performance of LLMs with SVD-LLM and Basis Sharing after LoRA
> fine-tuning to be similar by adjusting the number of ranks in both methods.
> The model used in the experiment is LLaMA-7B and the PPL is evaluated with WikiText-2.
> Under the PPL 7.77 after LoRA fine-tuning, SVD-LLM and Basis Sharing can achieve compression ratios of 20\% and 30\%, respectively. Basis Sharing can  generate 16 more tokens per second than SVD-LLM, which demonstrates the throughput enhancement under the same PPL.
>
> Under another PPL 9.14 after LoRA fine-tuning, SVD-LLM and Basis Sharing can achieve compression ratios of 30\% and 40\%, respectively. Basis Sharing can achieve 97 more tokens per second than SVD-LLM, which demonstrates the throughput enhancement under the same PPL.
>
> We will continue to explore the potential of Basis Sharing to enhance hardware performance in our follow-up work.
>
> **Reply to Weakness 2: Possibility of sharing $W_O$ across discontinuous layers**
>
> Thanks for your insightful comments. For $W_O$, it is not possible to realize 2-layer basis sharing even through discontinuous groups. The reason is that the Frobenius loss incurred by basis sharing across any two discontinuous groups is larger than that without basis sharing, as shown in Figure 4(b) in the paper on page 5. When considering sharing basis vectors across more than 2 discontinuous layers, there might be some potential in reducing the Frobenius loss since the upper bound of the Frobenius loss with basis sharing can be reduced, as shown in the **Case 1** in the answer to the second question raised by the last reviewer.
> Please find this analysis on the mathematical properties of matrices that can be shared across layers in the Appendix A.3 on page 15.
> To explore this sharing of discontinuous groups, we will develop efficient algorithms to group $W_O$ matrices of the layers whose basis sharing can reduce the Frobenius loss in our future work.
>
> **Answer to Question 1: Combination of Basis Sharing with pruning and quantization**
>
> Thanks for your question. In fact, we are working on combining Basis Sharing with pruning and quantization to further compress LLMs while maintaining their performance.
> To combine Basis Sharing with pruning,  we are working on pruning basis vectors and coefficient matrices with the state-of-the-art pruning techniques such as [1][2] where the impact of input activations is considered.
> Since basis vectors are shared across layers in Basis Sharing, to prune basis vectors, we will consider the impact of activations in
> all the shared layers to avoid the negative impact on LLM performance.
> The coefficient matrix unique to each layer can be pruned individually considering its activations. Similarly, those weight matrices that cannot be shared across layers can also be pruned.
>
> To combine Basis Sharing with quantization and inspired by [3], we are working on quantizing the values in the coefficient matrix unique to each layer to a power of 2 to further compress LLMs. Since basis vectors are shared across layers, quantizing them into a power of 2 might not be viable. Accordingly, we are exploring to quantize them into low bits as much as possible while still maintaining the performance of LLMs.
>
>
> [1] A Simple and Effective Pruning Approach for Large Language Models, ICLR, 2024
>
> [2] SparseGPT: Massive Language Models Can Be Accurately Pruned in One-Shot, ICML, 2023
>
> [3] ShiftAddLLM: Accelerating Pretrained LLMs via Post-Training Multiplication-Less Reparameterization, NeurIPS, 2024

---

> > ### Comment · Reviewer_NiaD · 2024-11-27
> >
> > Thank you for addressing the questions. I will keep my score to accept this paper. Best of luck!

---

### Official Review · Reviewer_zP6R · 2024-11-04

**Soundness:** 3
**Presentation:** 3
**Contribution:** 3
**Rating:** 5
**Confidence:** 4

**Summary:**

This work investigates the layer sharing of SVD bases across different layers and, based on these insights, proposes a new low-rank compression method. This method demonstrates improved performance compared to previous approaches under the same compression ratios. Plenty of ablation studies and end-to-end throughput are conducted to demonstrate the effectiveness of the proposed methods.

**Strengths:**

- The studies of basis sharing across different layers is interesting.

- The improvements shown in Table 1 is significant compared with previous baselines.

- End-to-end throughput are reported to demonstrates the effectivenss of the proposed methods.

**Weaknesses:**

- As shown in Table 1, while basis sharing can effectively enhance the performance of compressed models, some performance drop remains—such as a 0.04 decrease at 20% compression and a 0.07 decrease at 30%. Additionally, Figure 6 illustrates that only marginal throughput improvement is observed at a 20% compression ratio, which makes the proposed method less practical, as meaningful efficiency gains are accompanied by performance degradation.

- It would be good to show whether LoRA fine-tuning can recover the performance on zero-shot common-sense reasoning tasks, rather than perplexity.

- The evaluted models should contains more recent ones, like llama-3.1/3.2.

- The proposed method seems like a layer-sharing and SVD co-design methods for LLM compression. How does such method compared with direct apply layer-sharing and SVD compression, e.g., applying layer-sharing, then SVD or reversely.

**Questions:**

Is there any insights on why different layers would share the same SVD basis?

---

> ### Author Response · Authors · 2024-11-26
>
> Dear Reviewer zP6R,
>
> Many thanks for your suggestions and comments. We really appreciate your time and effort! To address your concerns, we have conducted further experiments as follows. We have also added the further results in the Appendix of the paper. We hope such experiments could clarify your doubts on the advantages of Basis Sharing.
>
> **Reply to Weakness 1: Balance between performance drop and efficiency gain**
>
> Thanks for your comments.
> Basis Sharing is our first step to verify the possibility of sharing parameters across layers in LLMs.
> To further improve the performance of LLMs under compression,
> we are working on
> integrating state-of-the-art techniques for performance enhancement into
>  Basis Sharing.
> First, different numbers of basis vectors can be allocated to different grouped layers according to their sensitivities to the performance of LLMs. As shown in [1], the non-uniform allocation of ranks in the traditional SVD decomposition can enhance the performance of LLMs.
> Second, when compressing weight matrices such as $W_K$ and $W_Q$, the resulting error of the attention $\sigma_r(XW_Q)^T\sigma_r(XW_K)$ instead of the separate compression errors needs to be considered. As pointed out in [2], considering the error of the attention result when compressing weight matrices can enhance the performance of LLMs significantly.
>
> [1] Adaptive Rank Selections for Low-Rank Approximation of Language Models, ACL, 2024
>
> [2] MoDeGPT: Modular Decomposition for Large Language Model Compression,  arXiv, 2024
>
> On the generation tasks,
> although the PPL degrades, the generated text with the compressed LLM model is still meaningful. The Table 13 in the Appendix A.7 on page 17 shows an example of the generated text under different compression ratios with LLaMA-7B.
>
> **Reply to Weakness 2: Performance enhancement on zero-shot common-sense reasoning tasks after LoRA fine-tuning**
>
> Thanks for your suggestions.
> We have executed LoRA fine-tuning to recover the performance on zero-shot common-sense reasoning tasks with LLaMA-7B. The results are shown in the following table, where the number within the bracket represents the accuracy without LoRA fine-tuning.  According to this table, the accuracy of zero-shot common-sense reasoning tasks can be enhanced by LoRA fine-tuning under different compression ratios. We have also added the result in Table 9 in the Appendix A.4 on page 16.
> | **Ratio** | **Openb.** | **ARC_e** | **WinoG.** | **HellaS.** | **ARC_c** | **PIQA** | **MathQA** | **Avg**      |
> |-----------|------------|-----------|------------|-------------|-----------|----------|------------|--------------|
> | **20%**   | 0.28 (0.28) | 0.67 (0.67) | 0.66 (0.66) | 0.49 (0.46) | 0.35 (0.36) | 0.72 (0.71) | 0.25 (0.25) | 0.49 (0.48) |
> | **30%**   | 0.28 (0.27) | 0.63 (0.63) | 0.64 (0.63) | 0.45 (0.40) | 0.32 (0.30) | 0.70 (0.68) | 0.25 (0.24) | 0.47 (0.45) |
> | **40%**   | 0.24 (0.22) | 0.54 (0.52) | 0.60 (0.61) | 0.40 (0.35) | 0.29 (0.27) | 0.66 (0.62) | 0.24 (0.23) | 0.42 (0.40) |
> | **50%**   | 0.22 (0.18) | 0.49 (0.42) | 0.59 (0.57) | 0.36 (0.31) | 0.24 (0.23) | 0.62 (0.58) | 0.22 (0.22) | 0.39 (0.36) |
>
> Table: The performance on zero-shot common-sense reasoning tasks using LLaMA-7B compressed with Basis Sharing, with and without LoRA fine-tuning. The number in the bracket is without LoRA fine-tuning.

---

> ### Author Response · Authors · 2024-11-26
>
> **Reply to Weakness 3: Performance evaluation with more recent LLMs**
>
> Thank you for your comments. Due to time limit, we applied Basis Sharing on LLaMA3.2-3B on some datasets. The results are shown in the following table and we have also added the result in Table 10 in the Appendix A.5 on page 16.
> According to this table,
> Basis Sharing still has a better performance than SVD-LLM. %Compared with the other model,
> However, LLaMA3.2-3B suffers more performance degradation compared with other LLMs when  Basis Sharing or SVD-LLM is used.
> This reason is that
> %But Basis Sharing suffers less than SVD-LLM.
> different from LLaMA-7B and LLaMA2-7B, group query attention (GQA) is used in LLaMA3.2-3B. Due to GQA,
> the dimension of $W_K$ and $W_V$ becomes
> $d_1>d_2$ where $d_1$ and $d_2$ are the number of rows and columns for such a matrix. Due to this mathematical property, the Frobenius
> loss of the matrix concatenating the two matrices horizontally in two layers tends to be larger than the sum of the Frobenius loss of each layer. Therefore, sharing basis vectors across two layers tends to bring larger Frobenius loss, so that such types of matrices will not be shared across layers. We have analyzed this properties in \textbf{Case 2, 3} of the second question raised by the last reviewer, as shown in the Appendix A.3 on page 14.
>
> To address the challenge described above, we are working on vertically concatenating those weight matrices across layers and decompose such concatenated matrices for basis sharing.
> With this vertical concatenation, the Frobenius loss incurred by basis sharing across layers can be reduced, as shown in the \textbf{Future work} in the second question raised by the last reviewer, which can be found in the Appendix A.3 on page 14-15. We will continue to work on this and enhance our method for such state-of-the-art LLMs.
> | Compression Ratio | Method               | WikiText-2 ↓ | Openb. | ARC_e | WinoG. | HellaS. | ARC_c | PIQA | MathQA | Average ↑ |
> |--------------------|----------------------|--------------|--------|-------|--------|---------|-------|------|--------|-----------|
> | 0%                | Original            | 7.84         | 0.31   | 0.75  | 0.70   | 0.55    | 0.42  | 0.77 | 0.35   | 0.55      |
> | 20%               | SVD-LLM             | 38.39        | 0.19   | 0.53  | 0.57   | 0.33    | 0.24  | 0.63 | 0.24   | 0.39      |
> |  20%                  | Basis Sharing       | 22.48        | 0.20   | 0.54  | 0.58   | 0.35    | 0.25  | 0.65 | 0.25   | 0.40      |
> | 30%               | SVD-LLM             | 44.22        | 0.14   | 0.41  | 0.54   | 0.30    | 0.19  | 0.59 | 0.23   | 0.34      |
> |  30%                  | Basis Sharing       | 27.41        | 0.15   | 0.44  | 0.56   | 0.30    | 0.20  | 0.59 | 0.23   | 0.35      |
> | 40%               | SVD-LLM             | 65.09        | 0.12   | 0.34  | 0.54   | 0.28    | 0.18  | 0.55 | 0.23   | 0.32      |
> |  40%                  | Basis Sharing       | 59.95        | 0.14   | 0.34  | 0.54   | 0.28    | 0.19  | 0.56 | 0.23   | 0.33      |
> | 50%               | SVD-LLM             | 106.42       | 0.12   | 0.31  | 0.51   | 0.27    | 0.18  | 0.54 | 0.22   | 0.30      |
> |  50%                  | Basis Sharing       | 104.69       | 0.12   | 0.31  | 0.49   | 0.27    | 0.19  | 0.54 | 0.23   | 0.30      |
>
> Table: Zero-shot performance of LLaMA-3.2B compressed using Basis Sharing and baselines
> under 20% to 50% compression ratios on WikiText-2 (measured by perplexity (↓)) and seven common-
> sense reasoning datasets (measured by both individual and average accuracy (↑))

---

> ### Author Response · Authors · 2024-11-26
>
> **Reply to Weakness 4: Comparison with direct layer sharing and SVD compression**
>
> Thanks for your comment. We have evaluated the two test cases you mentioned as follows:
>
> - **Test case 1**: sharing every two layers directly by restricting their weights to be the same:
>
>   In this case, we restricted every two consecutive layers to use the same set of weights by averaging their individual weights.  In this case, the compression ratio can reach 50\%. However, the PPL of LLaMA-7B  on WikiText-2 reaches 3.2e9 without fine-tuning, which might be unacceptable. If SVD was applied to compress weight matrices further, the performance of the model will degrade further.
>
> - **Test case 2**: using SVD to compress the weight matrix in each layer first and averaging their basis vectors for sharing between two consecutive layers:
>
>   In this case, we used SVD to decompose the weight matrix of each layer to achieve a compression ratio of 20\%. Under this compression ratio, we computed the average basis vectors of every two consecutive layers, denoted as $B_{avg}$. Afterwards, we restricted every two consecutive layers to use $B_{avg}$ as their basis vectors. In this case, the compression ratio can reach 29\%, but the PPL on WikiText-2 is 1.3e5 without fine-tuning, which might be unacceptable.
>
> **Answer to Question 1: Insights on basis sharing across layers**
>
> Thanks for your question. Basis Sharing in this paper was inspired by the identified similarity in the attention result and the output of adjacent layers in transformers in the previous work such as [1]-[4].
> For example, [1][2] observed a certain degree of similarity in the attention between adjacent layers. [3][4] pointed out that the outputs of two adjacent transformer layers are similar.
>
> Modern LLMs are based on transformer architecture. Therefore, we assumed that there are similarities in the attention and the output of adjacent layers in LLMs.
> The attention function and the output of a layer are realized by different weight matrices in a layer.
> Driven by this assumption, we have explored to allow  weight matrices across layers to share a set of basis vectors while still allowing different layers have unique coefficients to maintain their differences. This shared set of basis vectors implicitly indicates the similarity across layers identified in the previous work. The unique coefficients for different layers implicitly indicate that different layers still have their individual functions to maintain the performance of LLMs.
>
> [1] Leveraging Redundancy in Attention with reuse Transformers, arXiv, 2021
>
> [2] Sharing Attention Weights for Fast Transformer, IJCAI, 2019
>
> [3] The Unreasonable Ineffectiveness of the Deeper Layers, 	arXiv, 2024
>
> [4] What Matters in Transformers? Not All Attention is Needed,arXiv, 2024

---

> > ### Comment · Reviewer_zP6R · 2024-11-27
> >
> > Thank you for the responses. Most of my concerns have been addressed. However, I am still somewhat concerned about the practical utilization of this method, as the meaningful efficiency gains come at the cost of a non-negligible performance drop, not only ppl, but also for common-sense tasks. This performance degradation could potentially exacerbate issues such as hallucination or safety risks, which may not be fully captured by evaluating a single generated text.

---

> > > ### Author Response · Authors · 2024-12-01
> > >
> > > Thanks for getting back to us. We fully understand your concern. In fact, on SVD-based LLM compression, researchers are working on enhancing the performance of compressed LLMs by examining different techniques such as non-uniform rank selection methods, as suggested by Reviewer 1 8FuT in the reply. While state-of-the-art techniques focus on compression in one layer, Basis Sharing provides a fresh perspective to enhance the performance of compressed LLMs by enabling parameter sharing across layers. As the Reviewer 1 pointed out, this new perspective could be a substantial step for better compression of SVD-based methods.  To make this method more practical, we are actively working on the potential of combining Basis Sharing with state-of-the-art techniques such as non-uniform rank selection to improve the performance of LLMs. We will provide such results in our follow-up work in the near future.

---

### Official Review · Reviewer_8FuT · 2024-11-04

**Soundness:** 3
**Presentation:** 2
**Contribution:** 3
**Rating:** 8
**Confidence:** 4

**Summary:**

The paper introduces "Basis Sharing," a method for compressing Large Language Models (LLMs) by sharing basis vectors across different layers. This approach decomposes weight matrices into shared basis vectors and unique coefficients, aiming to reduce memory requirements.

**Strengths:**

1. The concept of concatenating weight matrices from multiple layers and sharing basis is interesting. It potentially increases the number of ranks to achieve a given compression rate.
2. The performance of the proposed method is better than previous methods on SVD like ASVD or SVD-LLM.

**Weaknesses:**

1. Seems the paper only considers the two-layer sharing case, it will be stronger if it can be expanded to multi-layers. The key drawback of the previous SVD method is that to achieve a 10% compression rate, for example, it requires pruning 55% of ranks. If more layers can be shared, this challenge can be mitigated to some extent.
2. The paper does not discuss how to allocate non-uniform ranks for SVD decomposition. Non-uniform allocation of ranks has been shown promising in recent works [1].
3. The final structure of the model is not presented in the paper. For example, what layers and what operations have been shared in the final compressed model?
4. Compared to pure SVD methods like SVD-LLM, a sharing basis can result in more memory costs or lower throughput since the proposed method can keep a higher rank and thus more active parameters at the inference time.

[1] Gao, Shangqian, et al. "Adaptive Rank Selections for Low-Rank Approximation of Language Models." Proceedings of the 2024 Conference of the North American Chapter of the Association for Computational Linguistics: Human Language Technologies (Volume 1: Long Papers). 2024.

**Questions:**

Please see the weakness.

---

> ### Author Response · Authors · 2024-11-26
>
> Dear reviewer 8FuT,
>
> Thank you for considering our work as both interesting and effective compared with other SVD-based approaches.
> We also appreciate your insightful suggestions on non-uniform allocation of ranks in different layers, which inspires us to further explore LLM compressions considering the sensitivities of different layers to the performance of LLMs.
> We have addressed your concerns as follows.
>
> **Reply to Weakness 1: Expansion of Basis Sharing to more than two layers**
>
> Thanks for your comments and your analysis. In the main body of the paper, we used two-layer sharing as an example. In the experiments, we have extended this sharing to multiple layers. The results are shown in Table 6 and Table 7 on page 9, where the performance of grouping every 2, 3, 4, 5, 6, 7, 8, 16 and 32 adjacent layers for basis sharing without and with LoRA fine-tuning is compared. According to these tables, under different compression ratios, the optimal setting of grouping adjacent layers is different. For example, in Table 6, under 20\% compression ratio, sharing every 5 adjacent layers is better. However, under 40\% or 50\% compression ratios, sharing every 2 adjacent layers is better.
>
> We agree with the reviewer on the analysis of the drawback of the state-of-the-art SVD methods. We aim to address the drawback of SVD methods to compress LLMs without affecting the performance of LLMs significantly. As you pointed out,
> when more layers are shared, to achieve the same compression ratio with that in SVD methods, basis sharing across layers can maintain a larger number of ranks and thus better performance.
>
> **Reply to Weakness 2: Uniform rank allocation for SVD decomposition**
>
> Thanks for your comments. After reading the paper on the non-uniform allocation of ranks for SVD decomposition, we found that this technique can be adapted to further enhance the performance of LLMs with Basis Sharing.
> As mentioned in [1], different layers can be allocated with different ranks according to their different sensitivities to the performance of LLMs. To determine the optimal ranks for each layer, an additional hypernetwork is used to balance the performance of LLMs and the rank of each layer.
> Inspired by this technique,
> the training technique with hypernetwork can also be used to determine the optimal number of basis vectors in different grouped layers for basis sharing.
> We will explore this in our follow-up work to further compress LLMs while maintaining their performance.
>
> **Reply to Weakness 3: Unclear final model structure**
>
>  Thanks for pointing out this lack of clarity. The final structure of two layers in LLaMA-7B with basis sharing is illustrated in the Figure 7 in Appendix A.1 on page 13, where
> $B_K, B_Q, B_V, B_{up}, B_{gate}$ are basis matrices shared in adjacent two layers.
> $C_K^{(i)}$, $C_Q^{(i)}$, $C_V^{(i)}$, $C_{up}^{(i)}$ and $C_{gate}^{(i)}$ are individual coefficient matrices unique for the $i$th layer. The multiplication of a basis matrix, e.g., $B_K$, and a coefficient matrix, e.g., $C_K^{(i)}$, corresponds to the original compressed weight matrix.
> %for $W_K, W_Q, W_V, W_{up}, W_{gate}$.
> $B_O^{(i)}, B_{down}^{(i)}$ are
> not shared across adjacent layers and they
> are individual basis matrices for the $i$th layer, which is equal to $U\Sigma$ after SVD decomposition on this layer. $C_{O}^{(i)}$ and $C_{down}^{(i)}$ are the corresponding $V^T$ after SVD decomposition on this layer.
>
> We have tested to share basis matrices above across different numbers of consecutive layers, e.g., 2, 3, 4, 5, 6, 7, 8, 16 and 32. The results are shown in Table 6 and Table 7 on page 9 in the paper.

---

> ### Author Response · Authors · 2024-11-26
>
> **Reply to Weakness 4: Memory cost and throughput of Basis Sharing in inference**
>
> Thanks for your comments. In our paper, we have demonstrated that under the same compression ratios the performance of LLMs with Basis Sharing is better than that with traditional SVD methods such as SVD-LLM. As you pointed out, under the same compression ratios, Basis Sharing can lead to a higher rank and thus more active parameters as well as a lower throughput than traditional SVD methods. However, such higher rank and lower throughput bring performance enhancement of LLMs compared with traditional SVD methods.
>
> To fairly compare the throughput and memory costs, we did further experiments. In these experiments, we maintained the performance of LLMs  with SVD-LLM and Basis Sharing after LoRA fine-tuning to be similar by adjusting the number of ranks in both methods. Afterwards, we compared the corresponding compression ratio and throughput of the two methods.
> The model used in the experiment is LLaMA-7B and the PPL is evaluated with WikiTex-2.
> Under the PPL 7.77 after LoRA fine-tuning, SVD-LLM and Basis Sharing can achieve compression ratios of 20\% and 30\%, respectively. Accordingly, Basis Sharing can achieve a lower memory cost under the same PPL. Besides, Basis Sharing can generate 16 more tokens per second than SVD-LLM, which demonstrates the throughput enhancement under the same PPL.
>
> Under another PPL 9.14 after LoRA fine-tuning, SVD-LLM and Basis Sharing can achieve compression ratios of 30\% and 40\%, respectively. Accordingly, Basis Sharing can achieve a lower memory cost under the same PPL. Besides, Basis Sharing can achieve 97 more tokens per second than SVD-LLM, which demonstrates the throughput enhancement under the same PPL.
>
> We will continue to explore the potential of Basis Sharing to enhance hardware performance in our follow-up work.

---

> > ### Comment · Reviewer_8FuT · 2024-11-29
> > **Response to Authors**
> >
> > I appreciate the authors for providing additional details and experiments on the proposed method. The authors have addressed most of my concerns, and I have increased my score to 8 to reflect stronger support for the paper's acceptance. The current version may still have minor issues related to efficiency and performance trade-offs, but the contribution of improving compression space for SVD-based methods is important. I believe this paper could be a substantial step for better compression space of SVD-based methods, and it can be further enhanced by non-uniform rank selection methods.

---

### Meta-Review · Area_Chair_ePiF · 2024-12-13

**Metareview:**

The paper introduces a novel method, "Basis Sharing," for compressing LLMs through cross-layer parameter sharing using singular value decomposition (SVD). This technique innovatively decomposes weight matrices into shared basis vectors and unique coefficients for each layer, enabling memory-efficient compression without significant performance degradation. Comprehensive experiments validate its efficacy, demonstrating improved performance over state-of-the-art SVD-based methods, particularly under high compression ratios. Additionally, the authors address critical concerns raised during the review process, providing expanded results, analyses, and detailed explanations of the method's underlying properties and practical applications.

**Additional Comments On Reviewer Discussion:**

Reviewers raised concerns about expanding basis sharing to more layers, addressing performance drops under high compression ratios, clarifying the final model structure, and comparing the method's hardware efficiency with traditional SVD-based techniques. The authors responded comprehensively, providing new experiments that demonstrated the method's scalability to multiple layers, fine-tuning enhancements for performance recovery, and insights into basis-sharing properties. They also added detailed evaluations of compression gains, memory cost, and throughput. These efforts effectively addressed key concerns

---

### Decision · Program_Chairs · 2025-01-22

Accept (Poster)